# Basecalling Using Joint Raw and Event Nanopore Data Sequence-to-Sequence Processing

**DOI:** 10.3390/s22062275

**Published:** 2022-03-15

**Authors:** Adam Napieralski, Robert Nowak

**Affiliations:** Institute of Computer Science, Faculty of Electronics and Information Technology, Warsaw University of Technology, 00-665 Warsaw, Poland; robert.nowak@pw.edu.pl

**Keywords:** basecalling, sequence-to-sequence, nanopore, bioinformatics, encoder decoder, attention, neural network, machine learning, joint processing

## Abstract

Third-generation DNA sequencers provided by Oxford Nanopore Technologies (ONT) produce a series of samples of an electrical current in the nanopore. Such a time series is used to detect the sequence of nucleotides. The task of translation of current values into nucleotide symbols is called basecalling. Various solutions for basecalling have already been proposed. The earlier ones were based on Hidden Markov Models, but the best ones use neural networks or other machine learning models. Unfortunately, achieved accuracy scores are still lower than competitive sequencing techniques, like Illumina’s. Basecallers differ in the input data type—currently, most of them work on a raw data straight from the sequencer (time series of current). Still, the approach of using event data is also explored. Event data is obtained by preprocessing of raw data and dividing it into segments described by several features computed from raw data values within each segment. We propose a novel basecaller that uses joint processing of raw and event data. We define basecalling as a sequence-to-sequence translation, and we use a machine learning model based on an encoder–decoder architecture of recurrent neural networks. Our model incorporates twin encoders and an attention mechanism. We tested our solution on simulated and real datasets. We compare the full model accuracy results with its components: processing only raw or event data. We compare our solution with the existing ONT basecaller—Guppy. Results of numerical experiments show that joint raw and event data processing provides better basecalling accuracy than processing each data type separately. We implement an application called Ravvent, freely available under MIT licence.

## 1. Introduction

DNA and RNA sequencing is an important tool in biological and medical research. DNA sequencing is essential in genomics, RNA sequencing is essential in transcriptomics. Considerable progress has been made since the first method for identifying nucleotides in DNA strands was proposed in 1977 [1]. The development of new technologies and sensors has led to the creation of increasingly better generations of sequencers, with regard to quality of results, volume of genetic material able to be sequenced in each trial, time savings, and cost efficiency [2].

Current progress in sequencing is being achieved thanks to new algorithms. Nanopore sequencing, a third-generation sequencing method, available since 2010, has as its key feature the ability to generate very long reads, tens or hundreds of times longer than other competitive solutions [3,4]. One of the most popular providers of devices based on this type of technology is Oxford Nanopore Technologies (ONT) [5]. Nanopore sequencing became a very popular sequencing technique after ONT launched its first MinION device [6].

The ONT nanopore sequencing process is based on measuring ionic current in the nanopore when DNA/RNA strand is passed through it. The currents’ values are then translated to the sequence of bases, a task known as basecalling. It can be expressed as the function described in Equation (Equation 1) and is a sequence-to-sequence processing task. A schematic illustration of sequencing and basecalling is presented in Figure 1.
(1)$$\begin{array}{ccc} F(X)=Y\\ \; & \mathrm{where}\\ \; & \; & X=\left\{x_{1},x_{2},...,x_{n}\right\},x_{i}\in R_{\geq 0}\\ \; & \; & Y=\left\{y_{1},y_{2},...,y_{m}\right\},y_{j}\in {\{}^{\prime }A^{\prime }{,}^{\prime }C^{\prime }{,}^{\prime }G{,}^{\prime }T^{\prime }\}\\ \; & \; & m<n\;(typically\;n\approx 8m) \end{array}$$

Basecalling is an essential step, mainly responsible for the output of the sequencer and the bases it reads. The software tools identifying bases from ionic current measurements are called basecallers. After the analysis of numerous published solutions, we devised their division into four categories based on two main factors:algorithms in use: Hidden Markov Models (HMM) vs. various deep learning models (including Recurrent or Convolutional Neural Networks);input data type: raw current signal data vs. events data (obtained through segmentation of raw current signal data).

The earliest solutions, such as Nanocall created by David et al. [7], or the first official ONT basecaller, Metrichor [8], used HMM models together with the Viterbi algorithm to decode the most probable sequence of k-mers that emitted observed events data sequence. Soon, deep learning models started to generate a lot of interest and began to be applied to other solutions. DeepNano, proposed by Boža et al. [9], and another ONT basecaller, Nanonet, were based on Recurrent Neural Networks and achieved better accuracy scores than previous ones, progressing from around 60% to approximately 80% [4]. However, they operated on event data, the processing of which began to be considered error-prone, creating inaccuracies which could then be propagated in the second stages of a pipeline, i.e., processing of events and deriving base sequences. Most later developments switched instead to single stage end-to-end processing of raw data. One of the first such tools used for this was BasecRAWller, proposed by Stoiber et al. [10], which changed basic statistical event detection on raw data to segmentation after RNN feature extraction. The next full end-to-end basecallers were Chiron [11], which used combinations of CNNs, RNNs, and CTC decoder. The current official ONT basecaller is called Guppy [12], which is closed-sourced and therefore its implementation details are not precisely known. The Mauler basecaller, proposed by Abbaszadegan [13], implemented a new approach using deep learning encoder–decoder with the attention model. Even more recent tools are URNano, based on U-net and proposed by Zhang et al. [14], and Causalcall [15], using TCN, which is a model gaining in popularity. Mentioned tools have increased the accuracy of single strand reads basecalling to over 90–95%. The latest state-of-the-art in terms of pure accuracy is Bonito [16], a research algorithm developed by ONT, and Heron, proposed by Boža et al., which can reach over 95% accuracy on some datasets [17]. Despite surpassing Guppy v3.3 by a large margin regarding accuracy, higher scores come at the cost of lower processing speed, which is most probably the reason why Guppy is still the official ONT basecaller. Nevertheless, all accuracy scores reached by nanopore basecallers are still lower than the over 99% scores of competitive short-read sequencers, like Illumina.

As for new conceptions in this field, both Abbaszadegan’s [13] and Zhang et al.’s [14] works mention that, apart from the direct use of raw data, the processing of its segmentation might be beneficial, providing more information to the model.

In this paper, we propose a novel basecaller called Ravvent, which performs joint sequence-to-sequence processing of raw and event data. Its name combines the words *raw* and *event*, relating to combination of raw and event data types. It is built as an encoder–decoder architecture of recurrent deep neural network. Our solution has an attention model and uses twin encoders for processing both input data types. It has been tested to verify how such joint processing influences the basecalling and its quality, and if this approach has potential in carrying out this task. The general idea of joint processing of different types of data is known in literature and presented, for example in Xiao’s et al. [18] and Chen’s et al. work [19]. However, to our knowledge, it has never been applied and examined as an approach to basecalling task.

Our paper is divided into five sections. Section 2 presents Materials and Methods used for developing and testing the proposed solution. Firstly, the data preprocessing pipeline, as well as datasets used—both simulated and real-life—are presented. Afterwards, an overview of the encoder–decoder models used in the developed tool is described, followed by details of its implementation. In Section 3, we show and discuss the results of the experiments conducted. Core ones are the tests focused on accuracy on simulated and real datasets, presented in Section 3.1 and Section 3.2, together with the speed analysis in Section 3.3. We also present an analysis of applied attention mechanism behaviour in Section 3.4. Our results were primarily compared to the Guppy v3.3.0 basecaller [20]. In Section 4, we discuss and summarize the results, comparing them with the initial hypothesis, indicating areas for future improvement and further research. We show that the joint processing of raw and event data reaches higher basecalling read accuracy than just using raw data and much higher accuracy than only carrying out event data processing.

## 2. Materials and Methods

### 2.1. Data Preprocessing

Sequences (time series) of raw current signal values, which are the plain output of the nanopore sequencers, were the primary data for basecalling. They were saved into fast5 files, a variant of Hierarchical Data Format 5 (HDF5), which enables an efficient data storage organised in a similar way to a file system [21]. For both training and evaluating a basecaller, the correct sequence of base symbols corresponding to raw sequences was required, serving as ground truth data. It included reference sequences for bases, stored in fasta files, and alignment data mapping each nucleotide base to a range of raw values. We used data sources described in Section 2.2 that already offered such alignment data, but in general, additional tools should be utilized to perform such mapping, e.g., Tombo [22] with its re-squiggle algorithm.

Time series with ground truth aligned data cannot be used directly for constructing a basecaller’s input data, since it is not available in typical sequencer output. We therefore propose the data preparation pipeline presented in Figure 2.

Raw signal values were used to detect events with an algorithm used in earlier versions of official ONT basecallers, specifically open-source Scrappie [23]. The algorithm is based on sliding window t-statistics and was applied in our solution with window length set between 6 and 9, and other parameters were set to default. Event detection provides segmentation of raw data into events, described by ranges of raw data values, but in contrast to ground truth alignment mapping, it does not use reference base sequence information, and such events are not guaranteed to correspond to bases. The event detection process itself tends to be error-prone, but it may still provide valuable information, in addition to plain raw data. In our algorithm, events have 5 features: (1) mean, (2) standard deviation, (3) length of raw data range, (4) difference of means between the previous and current event, and (5) mean squared.

Using raw and event data, together with ground truth aligned sequences, actual data samples were then created by gathering a maximal number of consecutive events where the corresponding number of raw data values included within them was not greater than the parameter $raw_{max}$. The next sample then started from the event being $event_{offset}$ positions after the initial event of the previous sample, enabling consecutive samples to overlap. $event_{offset}$ parameter acted as a sampling stride. Base sequence samples consisted of bases which corresponding raw values in the ground truth alignment data were included in the input raw samples.

Samples were then randomly shuffled and split into test, validation, and training datasets of sizes defined as fractions of the whole dataset. The next processing step was scaling, performed with standarization. Six separate scalers were used: 1 for raw data and 5 for each event data feature. Afterwards, raw and event data were padded and truncated to defined $raw_{max}$ and $event_{max}$ lengths.

Base sequences were put between *[START]* and *[END]* symbols, to mark out sequence spans, and then tokenized—each symbol being transformed to an integer token. This mapping dictionary consisted of 7 tokens: 4 nucleotides symbols, start, end, and padding symbols.

Finally, samples of raw, event, and reference token sequences were gathered into batches of size 128, forming data packages ready for processing by the model. During training, the model’s parameters were updated according to gradients computed after processing each batch.

All data used in our work was processed with defined parameters: $raw_{max}=200$ and $event_{max}=30$. These details, together with batch size, are marked on schematic examples of data samples in the Figure 3. The value of $raw_{max}$ was selected by us, with the assumption to cover wide enough span of raw data points and be short enough for performant RNN processing. $event_{max}$ was then set to a minimal number of events, always including all raw data points selected for a sample, based on the training part of real data. $event_{offset}$ was adjusted depending on a conducted analysis. For read accuracy evaluation, it was set to 6.

### 2.2. Datasets

#### 2.2.1. Simulated Sequences

Even though computerized data banks, especially those being a part of INSDC (International Nucleotide Sequence Database Collaboration), such as NCBI GenBank [24], are popular and actively used, full datasets containing raw output data files straight from ONT sequences are much less common and more difficult to obtain.

As a consequence, special simulators have been developed in order to instantly provide easily accessible and available raw electrical current signals data.

DeepSimulator was the first such tool, initially proposed in 2018 and updated in 2020 to provide better results [25,26]. Its core component is a pore model in the signal generation module, which uses BiLSTM-extended Deep Canonical Time Warping deep learning strategy. The simulator provides multiple parameters for tuning signal filtering and applied Gaussian noise to make it highly similar to real life data. Another recent tool is NanosigSim, proposed in 2020, which uses a processing pipeline similar to DeepSimulator, but operates on a BiGRU neural network instead of BiLSTM [27]. In their paper, the authors demonstrate that NanosigSim might offer slightly greater similarity to real raw sequences. However, since the differences are insignificant and DeepSimulator software is much better documented and used more widely, it was therefore used for data generation for simulated datasets described further. It was run in context-dependent mode with default filter and noise parameters.

To analyse the basecaller using data differing in complexity, 5 datasets were prepared. As a measure of DNA complexity, we used the linguistic sequence complexity introduced by Trifonov [28], in a variant proposed by Orlov et al. [29], with vocabulary restricted to words of size 6. Each dataset was based on two sequences of the same length, with one of them being used only for the training part, while the other was split into validation and test sets by 0.25 and 0.75 fractions, respectively. Their details are presented in Table 1.

The complexity of the datasets was proportional to the number of 6-mers appearing in the source base sequences. *K*-mers of length 6 were used, since 5–6 nucleotides are said to fit in a nanopore during single measurements, and should thus ensure that a reduction in the number of all 6-mers appearing results in a roughly proportional reduction of the complexity of raw current signal sequences in the context of the ability to learn and generalize from them. Basic 6-mers were chosen from a list composed of random permutation of all 6-mers, always starting from the first one—for example, Dataset 2 includes all 3 basic 6-mers of Dataset 1 plus 9 others. Each sequence in a dataset was a random variation with repetition of basic 6-mers, of defined length. A plot of all 6-mers appearing in the function of number of basic 6-mers, used for defining datasets’ characteristics, is presented in Figure 4. Lengths of sequences in datasets were set to roughly respect the proportion between the numbers of all 6-mers appearing across different datasets, especially with higher ids, in order to obtain fairer results from model trainings (by providing it with more data in the case of more complex datasets). Dataset 5, since it uses all of the 6-mers, represents not reduced and fully random bases sequences.

#### 2.2.2. Real-Life Sequences

Apart from the computerized bio-data banks mentioned earlier, data suitable for preparing new basecalling solutions is often publicly available as a supporting resource for articles on this topic. The dataset prepared and originally used for the development of the Chiron basecaller is one of the most popular pre-prepared datasets used in many other works [11,30]. The full dataset is split into training and evaluation parts, each of them consisting of 2000 reads of the *Escherichia coli* (*E. coli*) genome and 2000 reads of the *Lambda phage* genome. Our basecaller was trained on *Lambda phage* part, with evaluation part split by 0.15 and 0.85 into validation and test datasets, respectively. The same splitting was applied to *E. coli* dataset and its test part was additionally used for model evaluation.

### 2.3. Encoder–Decoder Models

Model architecture used in our proposed basecaller originates from the field of natural language processing translation, because basecalling is a sequence-to-sequence translation. We used encoder–decoder architecture of recurrent neural networks, initially proposed by Cho et al. [31] and Sutskever et al. [32]. We extended the model with an attention mechanism, which has been proven to greatly improve its translation capabilities, especially when dealing with longer sequences. We used multiplicative attention based on the one proposed by Luong et al. [33].

Figure 5 presents a general overview of the model structure. While processing raw data, the model consumes only raw data samples and passes them to raw encoder. Then, its output is used by attention mechanism and decoder, finally producing predicted bases sequence. Event data processing differs only in the two first steps, using event data samples and event decoder. For full joint processing, both data type samples and encoders are used in parallel. Then, their outputs are concatenated and passed to the attention mechanism and the decoder. Part of our architecture is described by Equations (Equation 2)–(Equation 4).

The encoder consumes an input sequence of length *S*—in this case, it is a sequence of raw or event data—and for each element of the sequence it produces an output, while passing information about its state for processing of next elements.

Decoder is provided with a *[START]* token to begin the decoding process. Then, at each time step, it produces output ($h_{dec}$) and state information.

An attention layer is used, queried by the decoder’s state over all encoder outputs (${\bar{h}}_{s}$). It is also provided with an input mask to explicitly ignore any encoder outputs corresponding to input padding. Prediction of the base at each time step is made using a beam search or greedy search decoding.

The attention layer first calculates attention weights $\alpha_{s}$ (Equation (Equation 2)) using specific score function (Equation (Equation 3)), with $W_{a}$ as a weight matrix shared over all time steps. Then, it computes a context vector *c* (Equation (Equation 4)), which is used in the decoder for the prediction of the probabilities of output sequence tokens at a certain time step. Therefore, attention allows the decoder to focus with different strengths on different parts of the input, which is applicable and needed in longer sequences of data processed by the basecaller.
(2)$$\alpha_{s}=\frac{\exp(score\left(h_{dec},{\bar{h}}_{s}\right)}{\sum_{s^{\prime }=1}^{S}(score\left(h_{dec},{\bar{h}}_{s}\right)}$$
(3)$$score\left(h_{dec},{\bar{h}}_{s}\right)=h_{dec}^{⊺}W_{a}{\bar{h}}_{s}$$
(4)$$c=\sum_{s}\alpha_{s}{\bar{h}}_{s}$$

Joint type is based on processing and encoding raw and event input data in parallel, using twin encoders, as shown in Figure 5. The outputs of these encoders are then concatenated on the first axis for each sample in a batch. The rest of the processing is executed in the same way as for raw and event components.

Both encoder and decoder are based on generalized Recurrent Neural Networks (RNNs). We applied one of the most popular type—i.e., Long-Short Term Memory (LSTM) network, with its bidirectional variant used in encoder (BiLSTM). Its output, to be used in attention mechanism, is the concatenation of LSTM cells’ outputs in both directions at the same step. During training, all models’ gradients were clipped by norm.

### 2.4. Loss and Accuracy Metrics

As a loss function, cross-entropy was used, being called over a sequence of the logits token predictions and a tokenized ground truth base sequence. It also respected masking on the ground truth sequence and did not take into account its padding. Loss was calculated on each data batch and averaged across all of the batches afterwards. It was monitored on training and validation datasets and logged after each computational epoch.

Two methods of accuracy measurements were used: subset [34] and read accuracy [35]. The first one, also called exact match accuracy, calculates a fraction of bases symbols identified exactly, also in the right position in the sequence. It was applied on the data samples. Read accuracy is a more significant method—it measures the identity of a whole predicted bases sequence relative to a reference once. To obtain the whole sequence, predicted samples are merged using pairwise Needleman–Wunsch algorithm. Using a minimap2 (v2.17) tool [36,37], we aligned basecalled reads to the reference bases sequence. Each read’s accuracy was defined as the number of matching bases in its alignment divided by the total alignment length including insertions and deletions, a.k.a. the ‘BLAST identity’ [35,38]. Accuracy from multiple reads was averaged using lengths of aligned blocks as weights.

### 2.5. Implementation

Implementation of our basecaller was done in Python using *tensorflow* and *keras* deep learning libraries [39]. The official *tensorflow* tutorial was used as a template for baseline code implementation [40]. For all inner RNNs (two encoders and a decoder), latent dimensions were set to 128. Encoders used 2 stacked BiLSTM layers, the decoder used 1 layer of LSTM. Gradients were clipped to ensure their L2-norm was less than 1. Scheduled sampling teacher forcing was applied during training, with a sampling ratio of 0.5. Core implementation files consisted of nearly 1500 lines of code (LOC) in total. The processing pipeline was tested on small data samples, focusing largely on the correct shapes of tensors passed between consecutive processing steps.

The software is publicly available on the project’s repository homepage [41], and a version used in analysis is presented on commit *7ab517e*. The detailed block diagram of Ravvent basecaller with selected network is presented in Figure 6.

## 3. Results

### 3.1. Basecalling Accuracy on Simulated Data

To analyse the behavior of our models on simulated datasets with different complexity, they were subjected to training and evaluation—the training took 100 epochs and applied $event_{offset}$ was equal to 1. A plot of read accuracy for each model using test datasets is presented in Figure 7, together with read accuracy of the ONT Guppy v3.3.0 basecaller subjected to the same evaluation sequences.

It can be seen that for the least complex datasets, accuracy scores for all types of own models are mostly above 90%—similar to the ONT Guppy that reaches around 92%, though performing at the same level for every dataset type. Interestingly, results on Dataset 1 of all own models are lower than on the two next, more complex, datasets. However, it is most probably due to *minimap2* tool not being adapted for that simple sequence. The event model gave the lowest accuracy for every dataset, but the gap between it and the other two increases dramatically for the more complex datasets, which confirms that plain event data is less usable for good quality of basecalling task. Joint and raw models’ accuracy also decreased with increasing complexity of datasets, but it did not drop below 90%, staying at a level competitive with Guppy. The joint model performed the best for all datasets, with an average gain of 1p.p over the plain raw model.

### 3.2. Basecalling Accuracy on Real Data

In addition to analysing simulated datasets, models were tested on real datasets. Their training was executed for 40 epochs on *Phage Lambda* dataset, $event_{offset}$ was equal to 6, due to its considerable size. Read accuracy achieved by each model variant and the ONT Guppy are listed in Table 2. They include scores achieved on *Phage Lambda* and *E. coli* datasets with Ravvent decoder’s beam width set to 5 and 1 (greedy search decoding).

A bigger beam width resulted in better performance—on average 0.86 p.p. for all models, but only 0.5 p.p. for better performing raw and joint models. However, speed of basecalling with bigger beam width was approximately 30% slower than with simpler greedy search decoding.

Joint model again performed the best in all tested cases, reaching on average 0.5 p.p. higher accuracy than the raw one. With bigger beam width, this advantage reached 0.73 p.p. The event model provided much lower and unsatisfactory scores.

Similar results of models trained on *Phage Lambda* on *E. coli* dataset confirmed that over-fitting to a specific organism type sequences did not take place.

The Guppy basecaller’s accuracy on real data is at approximately the same level as its accuracy on simulated datasets, which indicates its consistent performance and well-learnt ability to generalize over different datasets.

### 3.3. Basecalling Speed Analysis

Apart from the accuracy assessment, the speed of basecalling is also a relevant aspect to consider. To analyse the performance of the Ravvent, we run its trained models on the evaluation part of the *Phage Lambda* dataset 10 times, measuring and averaging the time it took to process all sequences. It was run with beam width set to 1. The final speed rate values were calculated by dividing the number of bases and the number of signal measurements in the sequences their processing time. The same process was conducted using Guppy v3.3.0. We run experiments on 4-core CPU machine (Intel family 6, model 61), with and without GPU NVIDIA GeForce GTX 1080 Ti enabled. Obtained results are presented in Table 3. Additionally, we attached speed rates declared in papers testing other basecallers, including Chiron and BasecRAWller from [11], Osprey and Bonito 0.3 from [17], and DeepNano-blitz 96 from [17], with approximate values scaled proportionally to the 4 CPU cores used by us. Since values in the referenced papers were not given for all units nor environment configurations that we examined, we filled the missing values with NA symbol.

The results confirm that the use of joint raw and event data processing impacts the basecalling speed, which is then 11% smaller than in raw only case and around 67% smaller than in event only case. Guppy’s speed on CPU only is 72% higher than Ravvent with joint model, but with GPU it reaches 86%. Chiron and BasecRAWller are the tools with the lowest speed, the reason is most probably the fact that they were developed as one of the earliest tools based on neural networks. On the contrary, DeepNano-blitz and Osprey have been published very recently and are deeply optimised to reach one of the highest speeds currently available. Bonito [16], as the newest ONT developed tool, is also able to process sequences with much higher speed. At this stage of the development, Ravvent has not been optimised regarding speed in most of its implementation. While it is very unlikely to reach processing rates of the fastest mentioned tools, some improvements in the speed through implementation changes are achievable, including parallelization or more efficient tensors usage.

### 3.4. Attention Mechanism

To review how the attention mechanism implemented performed, all types of trained models were run on suitable input data sequences and attention weights applied by models during such predictions were plotted in the form of a heat map—examples of which are presented in Figure 8. Points on the charts’ images indicated which encoder outputs (corresponding to the input sequence) were attended to for prediction of each base in the output sequence. The brighter the color, the bigger the weight assigned to a specific encoder output.

It can be seen that for both event and raw types, the points representing higher weights form a diagonal line on the plot. On event one, it is much more condensed, i.e., for single output base id there are usually 3 encoder outputs attended to. In the case of raw data, each predicted output base is largely based on a wider span of consecutive encoder outputs. As one of the characteristics of raw nanopore data is that a current is measured multiple times per each nucleotide and neighbouring nucleotides influence these measurements, it is assumed that correct identification of a base symbol requires the model to understand wider dependencies between raw data points. Attention weights plots indicate that our model demonstrates this property.

On the right-hand sides of both first plots, correct masking on the padding of the input data can also be observed as blank vertical zones.

As for the joint data, its attention weight plot corresponds to the way in which twin encoder outputs are processed in the model, i.e., concatenation. Therefore, there exist two distinguishable parts in the plot, representing separate raw and event plots, which are concatenated. This therefore confirms that such a model utilises information jointly from parts of raw and event data—for each predicted output base there are encoder outputs attended to in both parts.

## 4. Discussion

In this paper, we proposed a novel approach to basecalling, using joint raw and event data processing, and presented implementation of our own basecaller using this technique. The Ravvent basecaller is based on an encoder–decoder model with an attention mechanism. After evaluating it on simulated and real-life datasets, we confirmed that this approach shows potential for further exploration and development, since in all of the cases analysed it achieved accuracy scores that were higher than when processing only raw data, currently the most popular method.

We confirmed that plain event data processing is indeed much more erroneous, which was the reason for seeking new solutions based on raw data processing a few years back, one of the first ones being BasecRAWller [10]. However, when processed jointly with raw data, it appears to provide additional information that models can leverage to perform better.

Even though in terms of both accuracy scores and basecalling speed our basecaller achieved lower result values than state-of-the-art solutions, we find the presented approach promising, since other analysed tools are based only on raw data processing, while we proved that joint data types processing can result in increase of basecaller’s accuracy. We acknowledge that our model’s architecture was simpler than the ones of other tools, which is one of the reasons why we see a prospect of having its performance and results quality further increased—with code optimisation and especially with architecture development, including adjustments to RNN layers. Moreover, official ONT basecaller presumably has access to a large number of data sources, which undoubtedly enables it to learn and perform better. Our solution, for less complex (regarding appearing 6-mers) simulated datasets, performed considerably better than Guppy, which may additionally indicate the importance of amounts of training data, in which case the official solution has much greater training potential.

By proving that the joint processing performs better in terms of basecalling accuracy, we identify a practical value in such confirmation, which can be used as a clue for creating or developing basecallers with more complex architecture, optimised from the core to compete with the best available basecallers, by leveraging joint processing.

Lastly, in the neural machine translation field, more advanced versions of encoder–decoder models are currently gaining popularity, including transformer, which utilises a self-attention mechanism. Applying a joint processing approach to such model architecture might well boost its performance and provide more data for analysis of the approach itself.

## Figures and Tables

**Figure 1 sensors-22-02275-f001:**
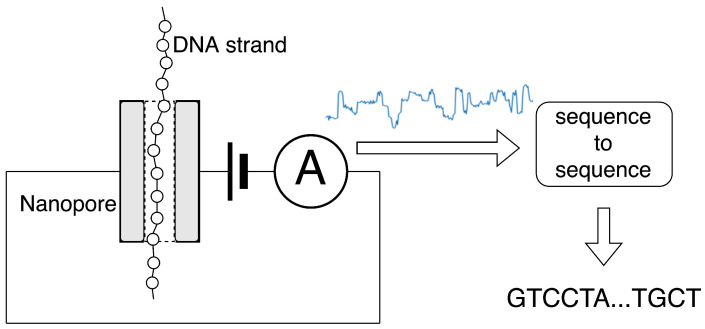
Illustration of the working principle of a nanopore DNA sequencer used in our research.

**Figure 2 sensors-22-02275-f002:**
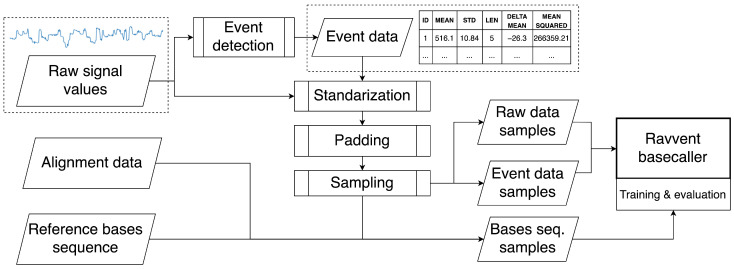
Data preparation pipeline. Our algorithm called *Ravvent* uses raw data, events, and ground-truth sequences.

**Figure 3 sensors-22-02275-f003:**
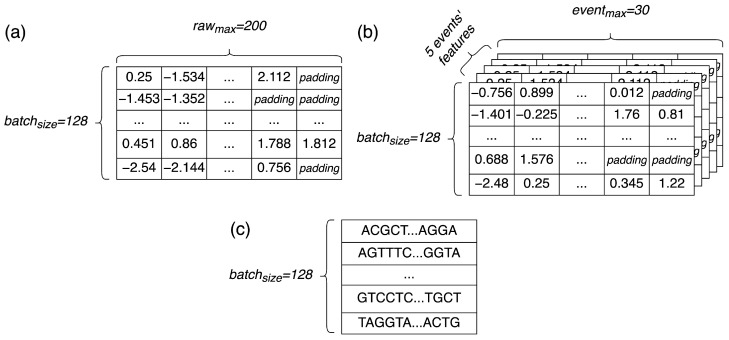
Schematic examples of data samples used by our model: (**a**) raw data, (**b**) event data, (**c**) bases sequences.

**Figure 4 sensors-22-02275-f004:**
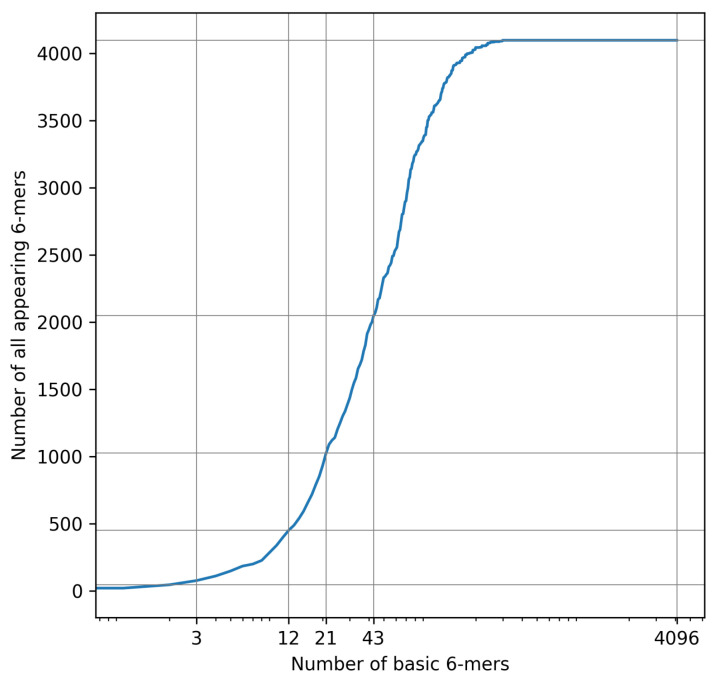
Number of all 6-mers appearing, depending on number of basic 6-mers.

**Figure 5 sensors-22-02275-f005:**
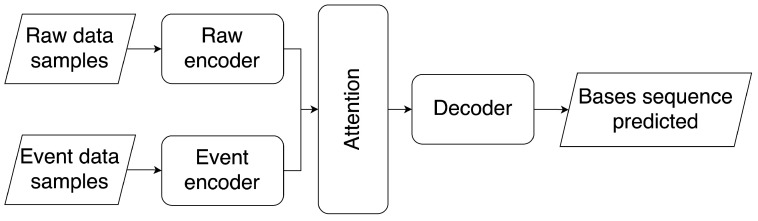
Schematic representation of model structure.

**Figure 6 sensors-22-02275-f006:**
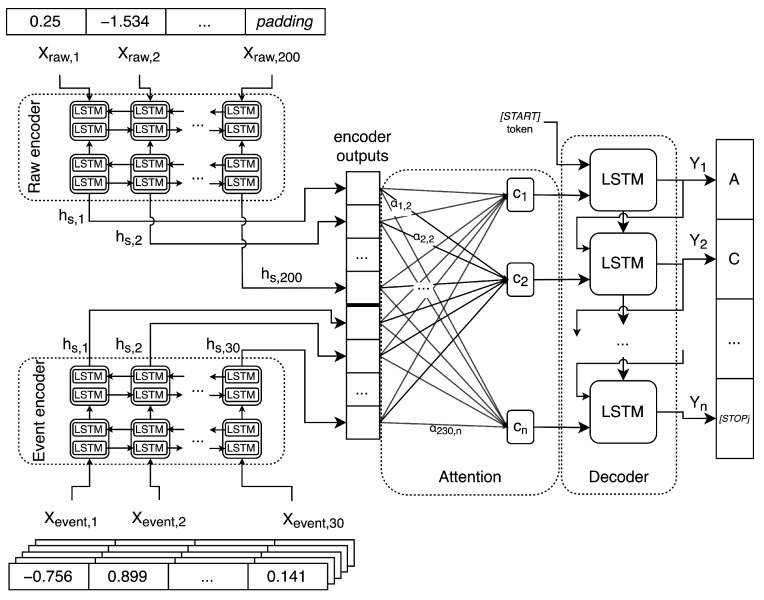
Detailed block diagram of Ravvent basecaller. It processes raw and event data with individual encoders using two BiLSTM layers. Their outputs are then subjected to attention mechanism. The produced context vectors *c* are then used by single LSTM layer decoder which generates the final base sequence.

**Figure 7 sensors-22-02275-f007:**
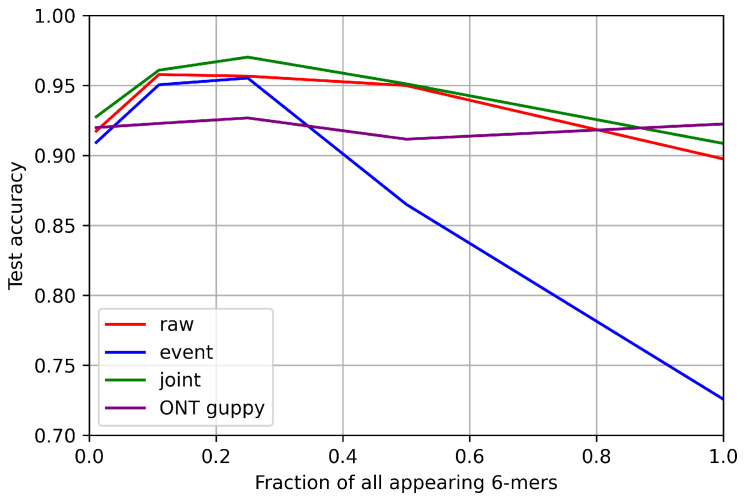
Test read accuracy of models trained on simulated datasets.

**Figure 8 sensors-22-02275-f008:**
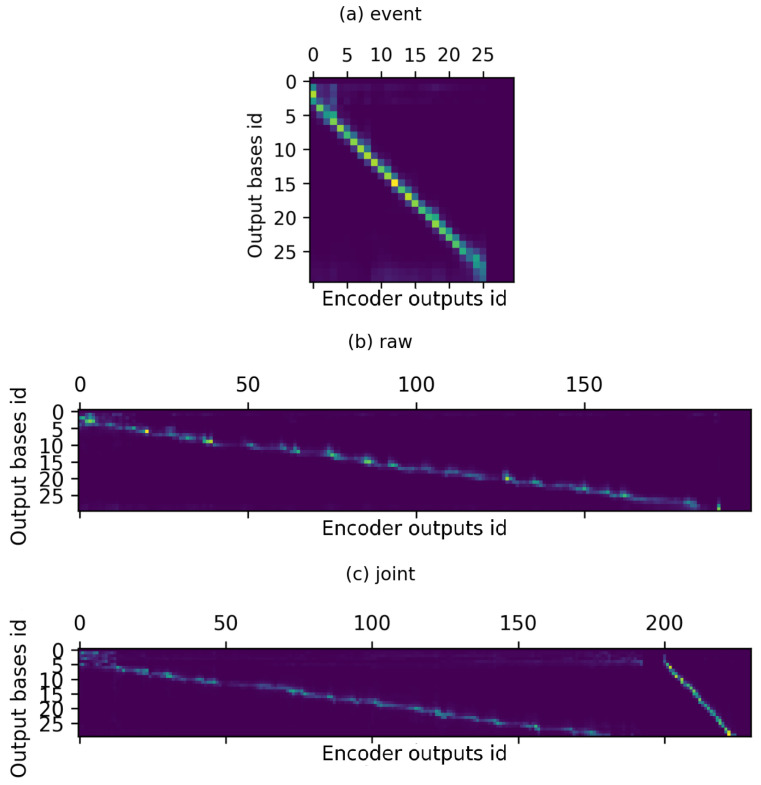
Plots of attention weights applied to different parts of input data depending on predicted output base id: (**a**) for event processing model, (**b**) raw processing model, (**c**) joint raw and event processing model.

**Table 1 sensors-22-02275-t001:** Details of reduced simulated datasets.

Dataset ID	No. of Basic 6-Mers Used	No. of All 6-Mers Appearing	Fraction of All 6-Mers Appearing	Sequence Length
1	3	45	0.01	25,000
2	12	450	0.11	75,000
3	21	1024	0.25	150,000
4	43	2048	0.50	300,000
5	4096	4096	1.00	600,000

**Table 2 sensors-22-02275-t002:** Test accuracy of models trained on real data.

Test Dataset	Beam Width	Ravvent			Guppy v3.3.0 hac
		Event	Raw	Joint	
*Phage Lambda*	1	70.015%	83.862%	84.199%	90.115%
	5	71.887%	84.296%	84.906%	
*E. coli*	1	69.901%	81.043%	81.267%	91.119%
	5	71.194%	81.154%	82.000%	

**Table 3 sensors-22-02275-t003:** Comparison of basecalling speed.

	Bases/s		Signals/s	
	CPU	GPU	CPU	GPU
Event (Ravvent)	3141	5933	26,047	49,194
Raw (Ravvent)	844	2802	7000	23,237
Joint (Ravvent)	747	2528	6193	20,962
Guppy v3.3.0 hac	2689	17674	22,686	149,083
Chiron	84	1652	NA	NA
BasecRAWller	324	NA	NA	NA
DeepNano-blitz 96	NA	NA	1.46 M	NA
Osprey	NA	NA	1.76 M	NA
Bonito 0.3	NA	NA	NA	370 K

## Data Availability

Publicly available datasets were analyzed in this study. Blueprints for simulated data can be found in project’s repository: https://github.com/adamnapieralski/ravvent-basecaller [41]. Real data of *Escherichia coli* and *Phage Lambda* can be downloaded from http://gigadb.org/dataset/100425 [11,30].

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
