# Peer review of "Basecalling Using Joint Raw and Event Nanopore Data Sequence-to-Sequence Processing"

_sensors, 2022, doi:10.3390/s22062275_

Round 1

Reviewer 1 Report

Comments to the Authors  

The manuscript “Basecalling using joint raw and event data sequence-to-sequence processing" by Adam & Nowak, present a new approach using event and raw data. The research results are plausible, and the manuscript is relevant. So, I suggest accepting the manuscript. 

I wish you good luck. 

Author Response

Dear Reviewer,

Thank you for considering our manuscript for publication and appreciating our work.

Yours sincerely,

Adam Napieralski
Robert Nowak

Reviewer 2 Report

The authors raise important problem of base calling in nanopore sequencing. The topic of the work fits to the Sensors journal scope. 
The manuscript is well-illustrated. It presents new software and should be published. I recommend adapt the text for wider readers' audience.
Advantage of the new tool is not proved, in fact. Yes, it has some potential, but it is not ready for practical application. Examples of real sequencing analysis are not enough. Nobody will sequence whole chromosome 1. Some large fragments with genome repeats and higher complexity present some interests, of course. The authors discuss some problems of basecalling for higher text complexity (in terms of n-mers in sequence), but it is very common estimate.. My point of view might be biased as biologist.
The manuscript could be published as a method paper.
So, I suggest to fix some technical points (minor revision).
First, add wording about ‘nanopore’ to the title. It might be not understandable to the readers in current variant. 
Line 2: ‘is used to compute’ - use word ‘define’ or ‘detect’, not ‘compute’ when talking about sequencing

Line 3: ‘, and this task is called basecalling’ - rephrase as a separate sentence. I’d suggest start from basecaling definition. Just my suggestion here.

Line 4: ‘HMMs’ - give the abbreviation in full in the Abstract.

Line 8: ‘segmentation into events’ - here is not clear what means ‘event’. The detailed explanation is in the main text. It refers to the method advantage, need write more clear in the Abstract.

Line 14: ‘ONT’ - need give abbreviation in full or avoid it in the Abstract.

Line 15: ‘confirm potential’ - it is weak conclusion. Need rephrase, show advantages of the method. Rewrite here.

Line 23: ‘...proposed in 1977.’ - need a reference here to the publication by 1977. 
I guess it is about Sanger sequencing, that is not topic of current work. However need cite correctly all the background publication.

Line 30: ‘The most popular..’  - avoid such phrases. It looks like an advertisement. May be ‘One of the popular’

Line 33: ‘...device.’ -  the sentence about Nanopore should have citation here.
‘This is used...’ - this sentence should be rewritten. It is not correct, and it has multiple, often not relevant citations.
For example about ‘the International Space Station’. Or need explain in more detail, that such sequencing is possible without gravitation, or what?
‘like Ebola or the recent SARS-CoV-2’ - too common sentence. The sequencing is important for multiple application, for sure. Rewrite the phrase, avoid duplicates in the references. 

Line 38: ‘DNA data storage [11–13]..’ - bulk citation - cite less, or add details about the cited references, usage DNA for data. 

Equation 1 - ‘m<n’ - need show typical sizes of m and n. Say, m (sequence size) is about 1000, what is n? Difference in order of magnitude?

Figure 1 - in the scheme ‘sequence 2 sequence’ - change digit ‘2’ to ‘to’. It has different sense - just 2 sequences or processing to.

Lines 49-52:  need add references

Line 54: ‘Metrichor’ - need reference.

Line 78: ‘Guppy is still the official ONT basecaller’ - this is opinion by the authors? What means ‘official’? Need a reference.

Line 81: ‘Abbaszadegan’s and Zhang’s et al. works’ - add references after each authors’ name.

Line 84: ‘Ravvent’ - could the authors comment why such name was chosen? It is an abbreviation, or mixture of some words relevant to the method?

Line 110: ‘fast5 files, a popular hdf5 format.’ - add reference to the format, comment about the format. It use word ‘popular’ need refer who define such popularity. Internet citations and web-links are acceptable too.

Figure 2 and figure 3 go together. I recommend separate them by the text. What one can see from Figure 2? Add comments below.

Why event(max)=30 ? (Figure 3)
Should it be always less then raw(max)=200
How the event size refer to sequence size (number of signals per DNA letter?)
Line 173: ‘logged after each epoch.’ - comment on ‘epoch’ as a computational epoch. It is kind of science jargon.

Line 174: ‘Two methods of accuracy measurements were used: subset and read accuracy.’ - need citations. I think more measures of accuracy were used for Illumina sequencing. It is worthy to cite.
‘subset’ accuracy seems to be accuracy of part of a sequence (word) or just order of sequence?
What specific errors for Nanopore  sequencing could be mentioned (errors in beginning of a sequence, errors in poly-A, skipping of some DNA basepairs)?

Figure 5 -
Add comments after the figure (1-2 sentences), what one can see there. Comment on the abbreviation used in the figure- LSTM, C1, C2...

Line 226: ‘American GenBank’ - give a reference. It is better call NCBI GenBank

Line 230: ‘weigh over 2TB’ - such example is artificial. Nobody uses such sequences. And it is not clear how file size was estimated - just from some data archive?

Line 248 and below: ‘The complexity ...’ - 
I’d recommend give definition of DNA sequence complexity, referring to previous papers.. Just search terms ‘DNA complexity’, ‘linguistic complexity’, ‘Lempel-Ziv complexity’. Theses publications are about 20 years old, but need cite such background works, explain why only 6-mers were used.

Figure 7. Legend ‘We used random, simulated data.’ - need comment on it not in the figure legend. Data size, simulation tool, is it uniform. Simulated sequence data are always differing from genome data in terms of genome repeat presence, and text complexity.

Add some explanation after the figure

Line 310: ‘approximately the same level’ - It is not visible from the table 2 - Guppy has better results.

Line 328: ‘Chiron and BasecRAWller’ - need references, citation, links.

Line 330: ‘DeepNano-blitz and Osprey’ - references to these tools?
Sane about ‘Bonito, as the newest ONT developed tool’

Line 369: ‘...a few years back’ - need reference.

Line 373: ‘we find the presented approach promising...’ - the advantages of the approach are not clear. Please explain why it is better or promising.

Reviewer 3 Report

In this study, the author developed an application called "Ravvent", which proposed a new method of processing raw data and event data together, unlike most base callers made directly from raw data. In this application, a machine learning model of encoder-decoder structure based on recurrent neural network is used, and this model contains two sets of encoders and an attention mechanism. The authors tested the application on simulated and real datasets and compared the accuracy of the tool in whole and in parts, as well as with ONT Guppy. However, there are still some issues:

  1. The advantages of the proposed base call approach over ONT Guppy don't show up. According to the results section of the article, several tests showed that Guppy performed better than Ravvent, and while the application performed better than base calls to raw or event data alone, it did not outperform its rival. Therefore, the practical value of this application needs to be examined.
  2. The new application presented in this article uses a model framework that is not particularly innovative. In fact, in this paper, the author also discussed the previous and current calculation models of base call methods, and the RNN and attention mechanism used in this paper are very common in machine learning models. This project only combines the two, and further structure or performance optimization is not shown.
  3. Some illustrations in this paper can be explained in more detail. For example, in Figure 3, the meanings of "rawmax", "eventmax" and "batchsize" are not explained in the notes. And what is the meaning of the number 5 in Figure (b)?
  4. The box line in Figure 4 needs to be improved as it does not represent the relationship between "raw", "event", and "joint" accurately.
  5. In Figure 5, the program diagram behind "Encoder Outputs" is drawn in a somewhat complicated way and can be simplified.
  6. In part2, the datasets also include data processing. But, the author wants to talk about the test datasets in part 2.5, which belongs to the materials part. Maybe this part can be modified and integrated with part 2.1 or it can be used as the beginning of part 2.
  7. In the second part of the article on Simulated sequences, DeepSimulator is mentioned as a good tool. However, it is not explained whether the simulation sequence in this paper is generated using this tool.
  8. In Table3, the lack of data in some tools suggests that the data table is not perfect. Is there a reason for that? Perhaps it should be explained.
  9. For (b) in Figure 8,the author thinks that the reason of “each predicted output base is largely based on a wider span of consecutive encoder outputs” is “require understanding wider dependencies between raw data points”. At this point, further explanation may be possible.

Round 2

Reviewer 3 Report

The authors have answered my questions.

This manuscript is a resubmission of an earlier submission. The following is a list of the peer review reports and author responses from that submission.

Round 1

Reviewer 1 Report

In this paper the authors claimed that a basecaller called Ravvent, which performs joint sequence-to-sequence processing of raw and event data, was proposed. Basecallers for nanopore are very important in translating the electrical signal into sequences for many biological or other novel applications, considering the nanopore model is quite complex to modelling. Their new basecaller is built as an encoder-decoder architecture of recurrent deep neural network. The solution had an attention model and uses twin encoders for processing both input data types. It was tested to verify how such joint processing influences the basecalling and its quality, and if this approach has potential in carrying out this task.

However, I have several suggestions.

(1) A detailed block diagram for this proposed method should be included to illustrated this new basecaller in summary. It can be expanded from Figure 4.

(2) basecallers for nanopore sequencing has promised more potential applications other than biological applications, for example DNA data storage. I think in the survey part of the nanopore sequencing devices, the non-biological application can be included to enrich the value of the basecallers with the nanorpore devices. See the references

[1] Molecular digital data storage using DNA. Nature Reviews Genetics, 2019

[2] DNA assembly for nanopore data storage readout. Nature Communications, 2019

[3] An artificial chromosome for data storage, National Science Review, 2021.

(3) In table II, why the accuracy value has so large gap to the Guppy basecaller should be explained further.

Reviewer 2 Report

I must say I like the idea of the work. The combination of using both event data and raw samples seem to be novel enough. From the explanation perspective, I believe the paper more-or-less covers all important aspects and questions about inner working of the basecaller. Obiously, there are many more details that would be nice to have but given the length constraints I feel quite ok with current descriptions. It is also very good that authors include reference to github (including commit) where the work can be checked.

However, I am not really convinced by the experiments. I completely understand that training modern basecallers takes weeks on modern NVidia hardware. This is however not an excuse for not training the base callers on larger amounts of data and for more epochs. As authors original results show in Table 2, the developed basecaller has underwhelming accuracy in practice (i.e. <40%). Given how sensitive is deep learning to datasets, I wouldn't want to therefore make any conclusions based on the results. This also means that I would like to see Ravvent being trained on multiple datasets (and there should be available ONT datasets of human, mice, some plants) to see whether the base caller isn't over-trained on a specific species.

Second comparison I miss is that of base-calling speed. Basecalling is a tradeoff between speed and accuracy - on one hand you have "realtime"-processing basecallers such as family of fast DeepNano basecallers (Blitz,Coral,Osprey) which provide reasonable accuracy with limited computed budged. On the other hand there are experimental high-accuracy basecallers (Bonito, Deepnano-Heron) which provide very high accuracy at the expense of speed. It would be very interesting to see where in this spectrum Ravvent lies, especially given unclear cost of its novel features (attention, twin-encoding).

Based on my glance over the basecaller source code, I have a feeling that the produced encoder is single layer of RNN. This is not very well explained in the paper but if this is true, it might be the potential reason for low accuracies on real datasets. Most of recent base callers use several layers (e.g. 4 RNN layers for Guppy, several CNN layers for Bonito). While attention & decoder layers may decrease the need for multiple layers, I am still not convinced that a single layer of encoder is enough to recognize complex-enough features before being fed to attention.

Minor comments:
- I believe the introduction misses the current state of the art in terms of pure accuracy - both Bonito basecaller and DeepNano Heron [https://arxiv.org/abs/2105.07520 preprint] push accuracies to over 95% on some datasets and surpass Guppy 3.3 by a large margin.
- Synthetic dataset exploration is interesting, however I am not sure how relevant it is to real-data results. On one hand, this is nice controlled environment for the experiment in Figure 7. On the other hand, it is hard to decide how this will apply to real data - Deep learning techniques tend to exploit any potential artefacts of simulators to the fullest. As an illustration, in case of GANs (where there is feedback cycle between simulator and discriminator) this process might even end up very wrong [https://techcrunch.com/2018/12/31/this-clever-ai-hid-data-from-its-creators-to-cheat-at-its-appointed-task/]
- Figure 6 - can you explain why raw accuracy is bigger than joint on your tests? This seems contradictory to the point of the paper. (I assume this is a consequence of just very limited training in section 3.1)
- Figure 7 should stress that datasets are simulated.
- I am not sure if I understand this correctly - do you use single-layer of BiLSTM as an encoder?

Overall, I am very torn with this paper. The idea is nice and well described and I would like to see it published. However the results are definitely not what I would expect to be reasonable. One thing is clear though, authors are honest with their claims and akcnowledge that this isn't a new state-of-the-art paper, rather a paper with interesting idea for future exploration.
I suggest that authors try some improved training procedure which tries to reach >~90% accuracy to really see whether this method really has a potential future. I also suggest a speed comparison to see how this fits into existing landscape of basecallers. 
